**Data Availability Statement:** All relevant data are within the paper and its Supporting Information files.

# Optimal paramedic numbers in resuscitation of patients with out-of-hospital cardiac arrest: A randomized controlled study in a simulation setting

**Bing Min Tsai**[1]**, Jen-Tang Sun**[2]**, Ming-Ju Hsieh**[3]**, Yu-You Lin**[1]**, Tsung-Chi Kao**[4]**, Lee-Wei Chen**[5,6,7]*****, Matthew Huei-Ming Ma**[3,8]**, Chiang Wen-Chu**[3,8]*

1 Division of Emergency Medical Service, New Taipei City Fire Department, New Taipei City, Taiwan,
2 Department of Emergency Medicine, Far Eastern Memorial Hospital, New Taipei City, Taiwan,
3 Department of Emergency Medicine, National Taiwan University Hospital, Taipei City, Taiwan, **4** Division Chief of Emergency Medical Service, New Taipei City Fire Department, New Taipei City, Taiwan, **5** Institute of Emergency and Critical Care Medicine, National Yang Ming University, Taipei City, Taiwan, **6** Department of Biological Sciences, National Sun Yat-Sen University, Kaohsiung, Taiwan, **7** Department of Surgery, Kaohsiung Veterans General Hospital, Kaohsiung, Taiwan, **8** Department of Emergency Medicine, National Taiwan University Hospital, Yun-Lin Branch, Douliu City, Taiwan

* drchiang.tw@gmail.com (WCC); lwchen@vghks.gov.tw (LWC)

## Abstract

### Background

The effect of paramedic crew size in the resuscitation of patients with out-of-hospital cardiac arrest (OHCA) remains inconclusive. We hypothesised that teams with a larger crew size have better resuscitation performance including chest compression fraction (CCF), advanced life support (ALS), and teamwork performance than those with a smaller crew size.

### Methods

We conducted a randomized controlled study in a simulation setting. A total of 140 paramedics from New Taipei City were obtained by stratified sampling and were randomly allocated to 35 teams with crew sizes of 2, 3, 4, 5, and 6 (i.e. 7 teams in every paramedic crew size). A scenario involving an OHCA patient who experienced ventricular fibrillation and was attached to a cardiopulmonary resuscitation (CPR) machine was simulated. The primary outcome was the overall CCF; the secondary outcomes were the CCF in manual CPR periods, time from the first dose of epinephrine until the accomplishment of intubation, and teamwork performance. Tasks affecting the hands-off time during CPR were also analysed.

### Results

In all 35 teams with crew sizes of 2, 3, 4, 5, and 6, the overall CCFs were 65.1%, 64.4%, 70.7%, 72.8%, and 71.5%, respectively (P = 0.148). Teams with a crew size of 5 (58.4%, 61.8%, 68.9%, 72.4%, and 68.7%, P<0.05) had higher CCF in manual CPR periods and better team dynamics. Time to the first dose of epinephrine was significantly shorter in teams

**Funding:** This study was funded by the Taiwan Ministry of Science and Technology (MOST 108-2314-B-002-130-MY3 and MOST 105-2314-B-002-200-MY3).

**Competing interests:** All authors have read and approved the final manuscript. There are no conflicts of interest to declare.

with 4 paramedics, while time to completion of intubation was shortest in teams with 6 paramedics. Troubleshooting of M-CPR machine decreased the hands-off time during resuscitation (39 s), with teams comprising 2 paramedics having the longest hands-off time (63s).

## Conclusion

Larger paramedic crew size (≧4 paramedics) did not significantly increase the overall CCF in OHCA resuscitation but showed higher CCF in manual CPR period before the setup of the CPR machine. A crew size of ≧4 paramedics can also shorten the time of ALS interventions, while teams with 5 paramedics will have the best teamwork performance. Paramedic teams with a smaller crew size should focus more on the quality of manual CPR, teamwork, and training how to troubleshoot a M-CPR machine.

## Introduction

### Background

Out-of-hospital cardiac arrest (OHCA) is one of the most critical situations that both paramedics and patients can experience in the prehospital setting. Over 176,100 cardiac patients were treated by emergency medical service (EMS) personnel in the United States annually.[1] In Taiwan, 9,815 OHCA cases per year were reported between 2000 and 2012, with an average survival rate of 9.8% at 180 days.[2] Recent studies found that chest compression fraction (CCF) during cardiopulmonary resuscitation (CPR) is strongly associated with survival rate, [3,4] and the widely adopted mechanical CPR (M-CPR) machine, which provides continuous chest compression, has the potential to increase the CCF during the entire course of resuscitation in the prehospital setting.[5] Previous studies also showed the different effects of several advanced life support (ALS) interventions, such as intubation or adrenaline (epinephrine) administration, on OHCA patients.[5–7] As the aforementioned interventions were performed by paramedics in the prehospital setting, the number of paramedics in the scene will certainly have a potential influence on the performance of these interventions.

### Importance

Paramedic crew size was supposed to be positively associated with the process and outcome of resuscitation of OHCA patients, but this viewpoint is still under debate.[8–12] Theoretically, the more manpower in the field, the earlier the interventions could be achieved, including regular handover to maintain high-quality CPR, higher CCF ratio, rapid defibrillation, administration of resuscitation drug, and better teamwork performance.[11–14] However, some studies showed controversial opinions. One simulation-based study reported no difference in CPR time between teams with 2 paramedics and those with 4 paramedics.[10] However, another retrospective study showed that a larger crew size of up to 7 paramedics was associated with higher survival to discharge rate.[12] Moreover, how the crew size affects the resuscitation processes including manual CPR quality, deployment of mechanical CPR machine, time to completion of advanced life support interventions, and teamwork dynamics also remained unclear. This issue is an important knowledge gap for OHCA resuscitation in the prehospital setting, policy-makers, and training curriculum in worldwide EMS.

### Goals of this investigation

To help determine the best crew size for resuscitation of an OHCA patient in the field before ambulance transportation, we designed a simulation-based randomised controlled trial with cardiac arrest scenario equipped with widely adopted mechanical CPR to test different crew sizes. We hypothesised that a larger crew would have better resuscitation performance including CCF, time to advanced life supports, and team dynamics than a smaller crew. The tasks that can possibly affect the hands-off time during CPR were also analysed.

## Methods

### Trial design study design and oversight

This study was a scenario-based randomised controlled trial held within the paramedic continuing education program of the New Taipei City Fire Department from September 1, 2018 to January 31, 2019. The study protocol was approved by ethics board of National Taiwan University Hospital.

### Participants study population and design

Among all 399 paramedics from all over New Taipei City Fire Department that attended continuous education program, 140 participants, selected through random sampling, were included in the trial. Those who were unwilling or physically unsuitable to attend this study were excluded.

### Sample size calculation

This trial was designed to detect an absolute difference of 25 percentage points in the CCF between the highest (90% estimated) and lowest (65% estimated) study groups based on the previous statistic data from department training, with the use of a two-sided alpha level of 0.05 and a power of 80%.

### Implementation and randomisation

First, we labelled the paramedics who were included in the continuous education program and then used stratified sampling based on the ratio of paramedics from every district of the city to select the initial 140 paramedics as study participants. After obtaining the informed consent from all participants, we collected their baseline data. Then, the participants were labelled again and were randomly allocated to five groups with different crew sizes (i.e. crew size groups of 2, 3, 4, 5, and 6). Every group contained 7 teams to be tested in the same scenario.

### Sequence generation and allocation concealment

All randomisation sequences were computer generated from a randomise sequence generating website.[15]

### Interventions and setting

After allocating all participants, all teams were given the same simulation scenario: "A 60-year-old man called 119 due to severe chest pain but eventually fainted. You and your crew were dispatched to the scene and saw that the patient is lying on the floor in his house, and no other persons appear to be nearby." The scenario was declared by a supervisor, and the supervisor would not have further communication with the crew until the whole test course is finished.

Paramedics here in Taiwan resuscitate the OHCA patients in the prehospital setting following the recommendations of ILCOR (International Liaison Committee on Resuscitation), but the timing of applying a mechanical CPR device during resuscitation has not been uniformly regulated in the EMS protocol of New Taipei City. Hence, the time for deployment of M-CPR device was decided by every team themselves.

In order to get a non-shaded view during our video recording, all crews were under the surveillance of three cameras simultaneously recorded from three different angles. The scenario environment and camera setup location are shown in Appendix A in S1 Appendix.

The manikin used was Laerdal Megacode Kelly, the defibrillator used was Zoll X Series, and the M-CPR device used was Physio-Control LUCAS.

## Blinding

Not applicable.

## Primary and secondary outcomes

The primary outcome of this study was overall CCF. Secondary outcomes were CCF during manual CPR period, time to advanced life support interventions (e.g. administration of first-dose epinephrine and accomplishment of endotracheal intubation), and teamwork performance.

Teamwork performance was measured using a pre-designed, structuralised evaluation form and rated on four aspects: leadership dedication, awareness and communication during changes of patient heart rhythm, closed-loop communication during medication administration, and awareness of unnecessary hands-off during CPR. Based on the final scores of every team, we graded their teamwork performance in three levels: good, average, and poor (Appendix B in S1 Appendix).

Both primary and secondary outcomes were analysed by reviewing the video recordings of all scenarios captured by two independent reviewers. Interobserver correlation coefficient was calculated and reported.

## Statistical analysis

We used Microsoft Excel 2007 to record the data, and IBM SPSS 22.0 was used to perform all statistical analyses. The descriptive statistics of the participants' demographic data are presented as count, mean (standard deviation), and percentage. The Kruskal-Wallis method was used to examine significance for sample size limitation (i.e. only 7 outcome data in each crew-size group), not fulfilling the assumption of normality. We also used the Kaplan-Meier method to compare the time-to-accomplishment of ALS intervention (i.e. first-dose of epinephrine and success of endotracheal intubation). For interobserver correlation coefficient, we used two-way random model of intraclass correlation coefficient reliability. A two-tailed P value of <0.05 were considered significant. Post hoc tests were performed to do pairwise comparisons for the primary endpoint (CCF) and Kaplan-Meier curves, with the adjusted level of the p-value for multiple comparisons.

## Results

### Participant flow and recruitment

During the study period, 140 of 399 paramedics from New Taipei City were enrolled through a stratified sampling process. The enrollees were further randomly allocated to five groups (i.e. comprising 2, 3, 4, 5, and 6 paramedics). The participant flow of the study is shown in Fig 1.

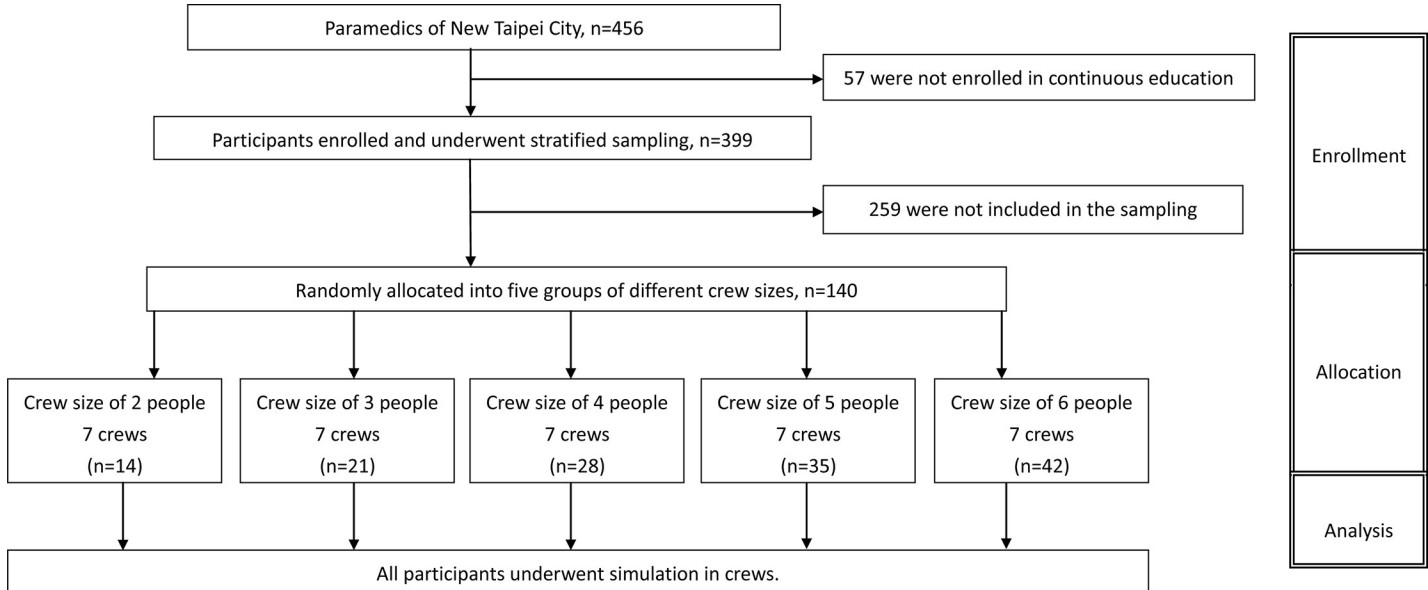

**Fig 1. Flow chart of the study process (following the CONSORT statement of randomized control trial).**

### Baseline data

The baseline demographic data of the participants are presented in Table 1, which showed that there was no difference between groups after the randomisation.

### Outcome and estimation

Of all 35 teams with 2, 3, 4, 5, and 6 paramedics, the overall CCF were 65.1%, 64.4%, 70.7%, 72.8%, and 71.5%, respectively (P = 0.148). The CCFs in manual CPR periods were significantly high in teams with 5 paramedics (58.4%, 61.8%, 68.9%, 72.4%, and 68.7%, P<0.05) (Table 2). The scatter plot of CCF data was shown in Appendix C and D in S1 Appendix. When examining the reason for the decrease in CCF (i.e. hands-off time during CPR), we found that the hands-off time was shortened as crew size increased. The time of chest compression and the causes of chest compression interruption in different teams were presented in Fig 2. The longest period of interruption was observed in teams with a crew size of 2 paramedics due to M-CPR device troubleshooting.

There was a significant difference in the time to first dose of epinephrine (316.29 s, 252.43 s, 179.91 s, 176.71 s, and 147.29 s; P = 0.004) and time to finish intubation (277.6 s, 212.0 s, 198.0 s, 196.9 s, and 130.1 s; P = 0.036) among the different crew sizes. The Kaplan-Meier test showed that the time to the first dose of epinephrine was significantly shortened in teams with a crew size over 4 paramedics (Fig 3A), while time to completion of intubation was shortest in teams with a crew size of 6 paramedics (Fig 3B). Teamwork performance rating was positively associated with increasing CCF (P = 0.01 for trend), and larger crew size was found to have better team dynamics (Fig 4). A crew size of 5 paramedics had the best teamwork performance as shown in Fig 4 (visual difference). The interobserver correlation coefficient reliability was 0.99 for the two video reviewers.

### Limitations

There are some limitations in this study. First, because the examined paramedics were randomly distributed to different teams, each resuscitation team was composed of unfamiliar

**Table 1. Demographic characteristics of the participants.**

| Demographic characteristics | | | | | | |
|---|---|---|---|---|---|---|
| | 2 people (n = 14) | 3 people (n = 21) | 4 people (n = 28) | 5 people (n = 35) | 6 people (n = 42) | P value |
| Age, year, mean (SD) | 34.1 (3.2) | 35.2 (3.3) | 32.0 (3.9) | 33.5 (3.7) | 33.0 (4.1) | 0.059 |
| EMT seniority, year, mean (SD) | 9.6 (2.3) | 11.3 (3.6) | 10.2 (3.3) | 10.7 (3.9) | 9.4 (2.8) | 0.192 |
| Paramedic seniority, year, mean (SD) | 5.6 (2.5) | 6.1 (3.2) | 5.6 (2.4) | 5.7 (3.1) | 5.0 (2.5) | 0.594 |
| Last ACLS certification | | | | | | 0.485 |
| >1 year, %(n) | 50.0 (7) | 52.4 (11) | 53.6 (15) | 42.9 (15) | 40.5 (17) | 0.787 |
| ≦1 year, %(n) | 28.6 (4) | 38.1 (8) | 42.9 (12) | 45.7 (16) | 54.8 (23) | 0.462 |
| Don't remember, %(n) | 21.4 (3) | 9.5 (2) | 3.6 (1) | 11.4 (4) | 4.8 (2) | 0.286 |
| OHCA experience in last 3 years | | | | | | 0.159 |
| >30 times, %(n) | 7.1 (1) | 14.3 (3) | 28.6 (8) | 25.7 (9) | 33.3 (14) | 0.247 |
| ≦30 times, %(n) | 64.3 (9) | 81.0 (17) | 57.1 (16) | 60.0 (21) | 61.9 (26) | 0.480 |
| Don't remember, %(n) | 28.6 (4) | 4.8 (1) | 14.3 (4) | 14.3 (5) | 4.8 (2) | 0.119 |
| Experience as leader at OHCA scene in last year | | | | | | 0.995 |
| >10 times, %(n) | 7.1 (1) | 9.5 (2) | 7.1 (2) | 5.7 (2) | 9.5 (4) | 0.974 |
| ≦10 times, %(n) | 85.7 (12) | 85.7 (18) | 89.3 (25) | 91.4 (32) | 88.1 (37) | 0.964 |
| Don't remember, %(n) | 7.1 (1) | 4.8 (1) | 3.6 (1) | 2.9 (1) | 2.4 (1) | 0.934 |
| Experience of Intubation at OHCA scene in the previous year | | | | | | 0.964 |
| >10 times, %(n) | 7.1 (1) | 4.8 (1) | 3.6 (1) | 2.9 (1) | 4.8 (2) | 0.886 |
| ≦10 times, %(n) | 85.7 (12) | 95.2 (20) | 92.9 (26) | 91.4 (32) | 92.9 (39) | 0.971 |
| Don't Remember, %(n) | 7.1 (1) | 0.0 (0) | 3.6 (1) | 5.7 (2) | 2.4 (1) | 0.748 |
| Experience of IV insertion at any scene in the previous year | | | | | | 0.162 |
| > 10 times, %(n) | 14.3 (2) | 23.8 (5) | 10.7 (3) | 11.4 (4) | 11.9 (5) | 0.678 |
| ≦10 times, %(n) | 71.4 (10) | 71.4 (15) | 85.7 (24) | 88.6 (31) | 88.1 (37) | 0.276 |
| Don't remember, %(n) | 14.3 (2) | 4.8 (1) | 3.6 (1) | 0.0 (0) | 0.0 (0) | 0.057 |
| Experience of LUCAS operating at OHCA scene in last year | | | | | | 0.111 |
| >10 times, %(n) | 14.3 (2) | 19.1 (4) | 0.0 (0) | 5.7 (2) | 14.3 (6) | 0.142 |
| ≦10 time, %(n) | 71.4 (10) | 76.2 (16) | 96.4 (27) | 82.9 (29) | 81.0 (34) | 0.224 |
| Don't remember, %(n) | 7.1 (1) | 0.0 (0) | 0.0 (0) | 0.0 (0) | 0.0 (0) | 0.060 |
| Other MCPR Device, %(n) | 7.1 (1) | 4.8 (1) | 3.6 (1) | 11.4 (4) | 4.8 (2) | 0.711 |

EMT, emergency medical technician; ACLS, advanced cardiovascular life support; OHCA, out-of-hospital cardiac arrest; MCPR, mechanical cardiopulmonary resuscitation; IV, Intravenous; LUCAS, Lund University Cardiopulmonary Assist System; SD, standard deviation

partners and EMTs from different levels, which was somewhat different from their daily clinical practice. Some measurement variables like CCF or teamwork performance may be suitable in that situation. Second, due to the lack of resources, CCF was measured by reviewing the video footage of two reviewers instead of using a computerised program. However, we checked the interobserver consistency to guarantee the validity of the evaluation. By reviewing the video footage, the researchers were able to determine the CCF from the time the paramedic approached the patient; by contrast, if a computerised program was used, it can only be operated once the machine has been completely set up. Third, the resuscitation in this simulation study may not perfectly match the practice in the true prehospital setting, which might have some uncertainties that may interfere the crew such as the confined space or opinions from the families. Hence, further clinical research is needed to verify the findings in this study. Finally, this study examined the optimal crew size based on the CCF and prehospital ALS (i.e.

**Table 2. Primary and secondary outcome results.**

| | 2 people | 3 people | 4 people | 5 people | 6 people | P value | post hoc tests |
|---|---|---|---|---|---|---|---|
| | Primary outcome | | | | | | |
| CCF, %(SD) | 65.1 (8.5) | 64.5 (5.1) | 70.7 (6.7) | 72.8 (8.2) | 71.5 (9.5) | 0.148 | 2 = 3 = 4 = 5 = 6 |
| Hand compression Duration CCF, %(SD) | 58.4 (7.7) | 61.8 (10.4) | 68.91 (5.8) | 72.35 (6.7) | 68.73 (14.9) | 0.043* | 5>2 = 3 = 4 = 6 |
| MCPR duration CCF, %(SD) | 67.0 (19.6) | 75.5 (19.7) | 68.5 (14.5) | 70.1 (13.6) | 72.3 (8.2) | 0.883 | 2 = 3 = 4 = 5 = 6 |
| | Secondary outcome | | | | | | |
| Contact to recognition, sec (SD) | 12.9 (3.1) | 12.9 (4.9) | 12.6 (4.00) | 14.3 (3.2) | 17.0 (5.7) | 0.339 | 2 = 3 = 4 = 5 = 6 |
| Time without CPR, sec (SD) | 199.0 (59.3) | 257.7 (43.6) | 120.1 (35.1) | 118.3 (29.9) | 101.0 (34.1) | 0.011* | 4 = 5 = 6>2 = 3 |
| Time with hand CPR, sec (SD) | 205.0 (134.5) | 184.9 (94.5) | 180.0 (58.3) | 190.0 (101.0) | 187.1 (127.8) | 0.987 | 2 = 3 = 4 = 5 = 6 |
| Time with mechanical CPR, sec (SD) | 165.9 (132.0) | 102.4 (82.5) | 109.4 (59.7) | 153.9 (133.3) | 88.0 (79.0) | 0.641 | 2 = 3 = 4 = 5 = 6 |
| Time to finish, sec (SD) | 569.9 (101.2) | 445.0 (100.1) | 409.6 (71.0) | 461.6 (123.5) | 376.1 (87.2) | 0.031* | 3 = 4 = 5 = 6>2 |
| Mean hand CPR depth, cm (SD) | 3.8 (0.6) | 3.9 (0.6) | 4.0 (0.3) | 4.2 (0.6) | 3.9 (0.6) | 0.688 | 2 = 3 = 4 = 5 = 6 |
| Mean hand CPR compression rate, cpm (SD) | 108.6 (9.1) | 110.4 (10.6) | 104.1 (5.5) | 105.4 (4.1) | 111.4 (7.2) | 0.437 | 2 = 3 = 4 = 5 = 6 |
| Initialize M-CPR assembly, sec (SD) | 176.0 (156.8) | 235.3 (146.4) | 176.3 (81.4) | 204.7 (122.4) | 199.0 (144.1) | 0.938 | 2 = 3 = 4 = 5 = 6 |
| CPR interruption due to defibrillation, sec (SD) | 32.0 (23.5) | 30.4 (22.6) | 16.4 (8.6) | 11.9 (7.9) | 17.6 (10.1) | 0.200 | 2 = 3 = 4 = 5 = 6 |
| CPR interruption due to ventilation, sec (SD) | 43.5 (34.9) | 41.3 (30.5) | 21.7 (8.5) | 22.7 (6.8) | 18.1 (8.9) | 0.433 | 2 = 3 = 4 = 5 = 6 |
| CPR interruption due to placement of the M-CPR board, sec (SD) | 7.3 (3.9) | 5.0 (1.3) | 3.9 (1.0) | 5.8 (3.0) | 4.4 (1.9) | 0.294 | 2 = 3 = 4 = 5 = 6 |
| CPR interruption due toMCPR assembly, sec (SD) | 20.1 (8.4) | 17.4 (12.9) | 14.4 (8.8) | 12.4 (6.2) | 12.0 (9.3) | 0.424 | 2 = 3 = 4 = 5 = 6 |
| CPR interruption due to MCPR troubleshooting, sec (SD) | 62.7 (64.2) | 35.6 (40.2) | 32.9 (33.1) | 33.1 (24.1) | 29.0 (40.2) | 0.823 | 2 = 3 = 4 = 5 = 6 |
| CPR interruption for other reasons, sec(SD) | 35.5 (21.0) | 27.9 (14.1) | 31.4 (15.4) | 32.4 (17.6) | 27.0 (12.7) | 0.922 | 2 = 3 = 4 = 5 = 6 |
| Initiation of intubation, sec (SD) | 208.7 (62.5) | 158.1 (86.2) | 157.7 (74.3) | 162.6 (49.5) | 89.0 (71.3) | 0.103 | 6>2 = 3 = 4 = 5 |
| Completion of intubation, sec (SD) | 277.6 (90.2) | 212.0 (84.6) | 198.0 (77.3) | 196.9 (55.5) | 130.1 (68.0) | 0.036* | 3 = 4 = 5 = 6>2 |
| Administration of the first dose of epinephrine, sec (SD) | 316.3 (94.9) | 252.4 (92.5) | 179.9 (59.2) | 176.7 (50.5) | 147.3 (40.2) | 0.004* | 4 = 5 = 6>2 = 3 |
| Teamwork performance | | | | | | 0.010* | 5 = 6>2 = 3 = 4 |
| Good, n(%) | 0 (0) | 0 (0) | 1 (10.0) | 5 (50.0) | 4 (40.0) | 0.004* | |
| Medium, n(%) | 5 (26.3) | 4 (21.1) | 6 (31.6) | 2 (10.5) | 2 (10.5) | 0.118 | |
| Poor, n(%) | 2 (33.3) | 3 (50.0) | 0 (0) | 0 (0) | 1 (16.7) | 0.145 | |

MCPR, mechanical cardiopulmonary resuscitation; CCF, chest compression fraction; CPR, cardiopulmonary resuscitation; SD, standard deviation

endotracheal intubation and intravenous epinephrine). The results may not be applicable to those EMS systems without the recommendation of prehospital ALS in the field.

## Discussion

In this randomised controlled trial, we have three major findings. First, a larger paramedic crew size (≧4 paramedics) does not significantly increase the overall CCF in OHCA resuscitation, but has higher CCF in manual CPR period before the setup of the CPR machine. Second, a crew size of over 4 paramedics can also shorten the time of ALS interventions, but complex interventions (i.e. endotracheal intubation, compared with intravenous drug administration) require more paramedics in order to accomplish the interventions on time. Third, a crew size of 5 will have the best teamwork performance, and the teams with a smaller crew size should

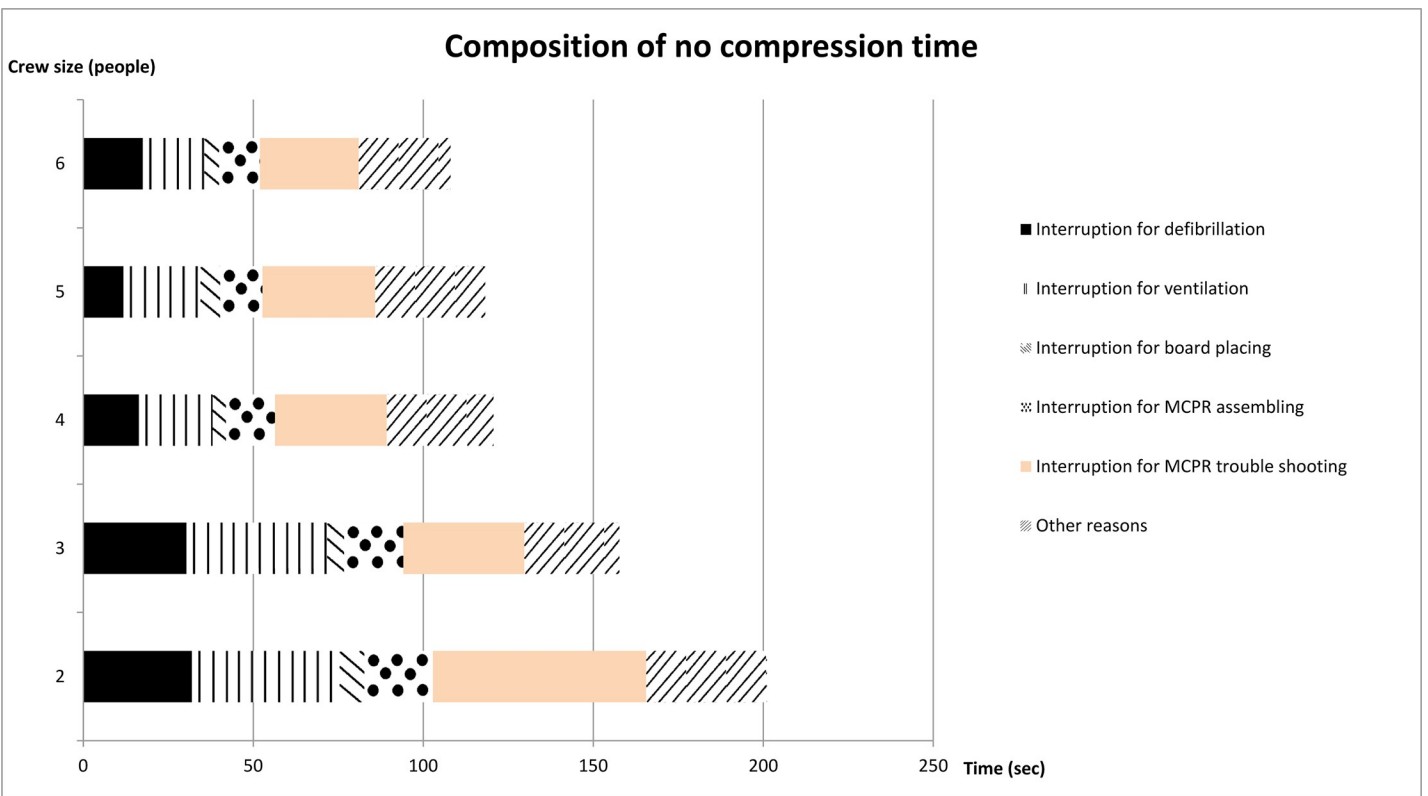

**Fig 2. Composition of no compression time.** * MCPR, mechanical cardiopulmonary resuscitation.

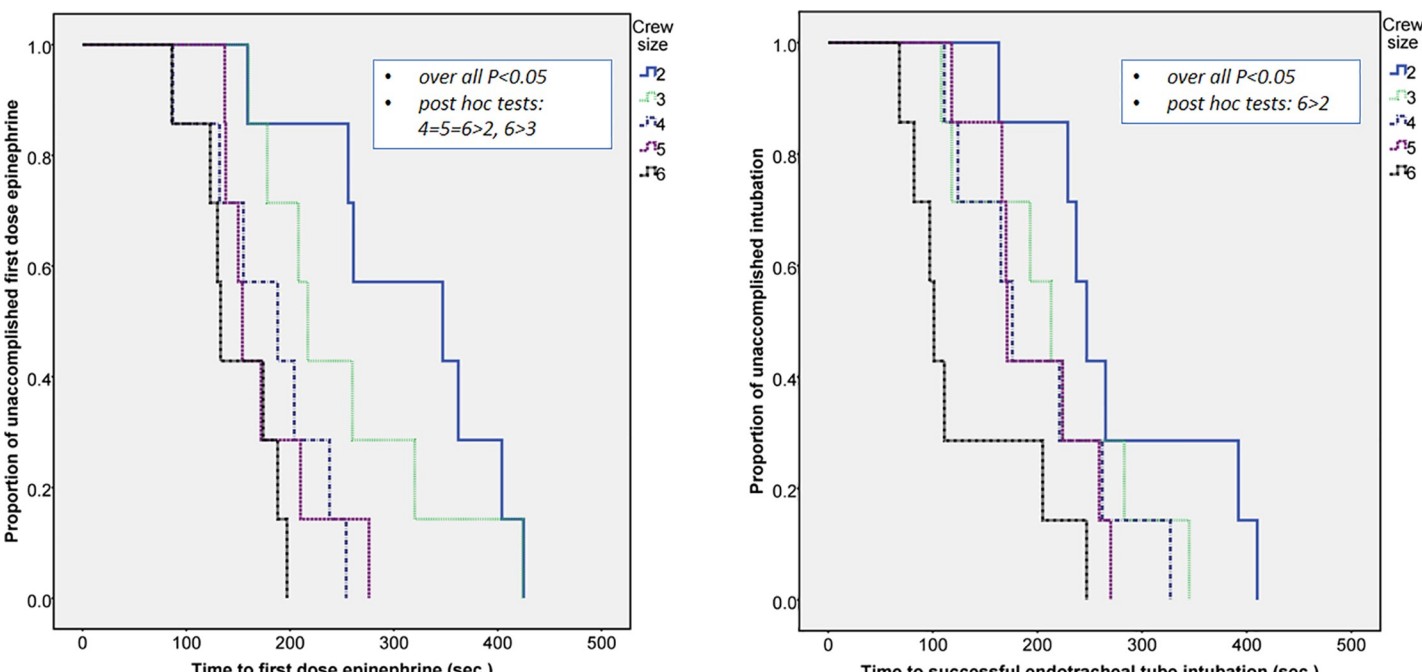

**Fig 3.** A. Accumulative successful rate of ALS interventions (intravenous epinephrine). B. Accumulative successful rate of ALS interventions (endotracheal intubation).

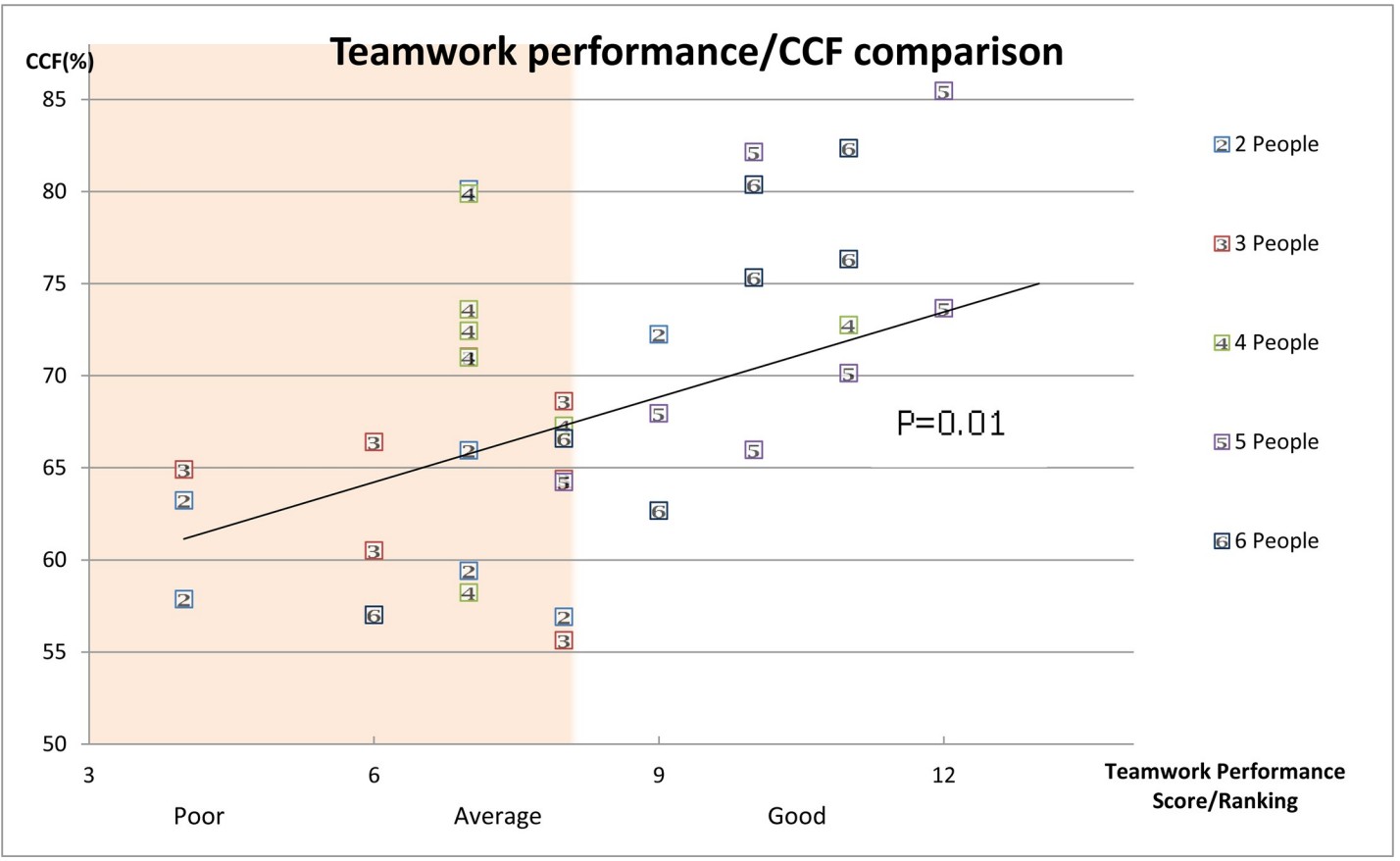

**Fig 4. Association between teamwork performance and chest compression fraction.** Larger crew size tends to have higher CCF and better teamwork performance rating. CCF, chest compression fraction.

focus more on the quality of manual CPR, on improving teamwork, and on training how to troubleshoot a mechanical CPR device.

Our study had some strengths compared with previous studies evaluating the number of paramedics during resuscitations. Martin-Gill et al. recruited 40 paramedics and divided them into teams with crew sizes of 2, 3, and 4 paramedics. They concluded that crew sizes had no significant impact on the "no flow fraction" during CPR.[10] However, the results of the current study were different from the findings of the previous study and informative in two aspects. First, we used the widely adopted M-CPR devices, which had not been tested in a previous study, in our scenario. Second, we divided the participants into teams with paramedic crew sizes of 2, 3, 4, 5, and 6, in which the larger crew size effect could be evaluated. Furthermore, our study provided complementary knowledge regarding the crew size issue. Sun et al. reported that a total crew size of 4 paramedics and a paramedic ratio of >50% had the best odds of survival among OHCA patients.[11] The reasons for the survival benefit in their retrospective study was supposed to be related to the provision of early ALS interventions, since literatures from Taiwan EMS suggested that epinephrine can improve survival to discharge rate for OHCA patients and prehospital endotracheal intubation can increase the odds of survival to discharge.[6, 7] Our study demonstrated that larger crew sizes contributed to the early accomplishment of ALS interventions. A crew size of over 4 paramedics can significantly shorten the time of intravenous epinephrine interventions, while a complex intervention such

as endotracheal intubation might require more paramedics (i.e. a 6-paramedic team) to accomplish the task within a short time.

The association between crew sizes and the CPR performance was deeply explored in this study. With regard to the reducible hands-off time during CPR, we found that hands-off time during CPR had dropped sharply as the crew size increased (trend P = 0.011). The most time-consuming event was the troubleshootingof the M-CPR device. Although without significance, we found that the interruption time of two people was almost twice as long as that of three people. By video review, we found that all crew members immediately noticed that the alarm on the M-CPR device malfunctioned, which increased their level of awareness regarding the situation; after the M-CPR was implanted, each crew member was simultaneously executing further interventions including endotracheal tube intubation and medication administration. Most of the time, the M-CPR device malfunctioned while all members were preoccupied by other procedures. Hence, although all members were aware that the device malfunctioned, without good teamwork and coordination, they will not be able to handle the incident immediately. Therefore, we suggest that teams with a smaller crew size should focus more on the quality of manual CPR, on improving teamwork, and on training how to troubleshoot a mechanical CPR device.

In this study, we found that a crew size of 5 paramedics had the best performance in resuscitation as all of them have a clearly defined task in the OHCA setting: one leader, one assigned in managing the airway, two as interchangeable chest compressor and defibrillator operator, and one assigned in establishing an intravenous access and administering medications. The team performance rating of a crew size of 6 paramedics was not as good as that of a crew size of 5 paramedics, and this might be due to the overcrowding in the confined space of the OHCA setting as well as due to ineffective communication caused by unclear task allocation as observed in the video review. Indeed, it has been mentioned in some teamwork trainer books that an ideal crew size should not be more than 6.[16, 17] However, we also observed that complex interventions like endotracheal intubation may require more paramedics in order to accomplish the task on time; however, given the said circumstance, only two paramedics are usually needed for airway management. Therefore, the number of paramedics who should be dispatched to the scene of a cardiac arrest should depend on who have been authorised by the local EMS to perform the interventions.

In summary, larger paramedic crew size (≧4 paramedics) does not significantly increase the overall CCF in OHCA resuscitation but have higher CCF in manual CPR period before the setup of the CPR machine. A paramedic crew size of over 4 can also shorten the time of ALS interventions, while a paramedic crew size of 5 will have the best teamwork performance. Paramedic teams with a smaller crew size should focus more on the quality of manual CPR, on improving teamwork, and training how to troubleshoot a mechanical CPR device.

## Supporting information

**S1 Appendix.**
(DOCX)

## Acknowledgments

We are very grateful to several paramedics in the department including Sun-Chiu Yang, Xue-Wei Lu, and Chun-Chieh Wang for their supervision. We would also like to acknowledge Mr. Marvin Chiu of Zoll Medical Taiwan and Ms. Chiao-Ling Zhan of Adison Bio-Medical Company for providing the Zoll X Series defibrillator and LUCAS 2 mechanical cardiopulmonary

resuscitation devices. We would like to thank all paramedics from New Taipei City Fire Department, who are very willing to help and participate in this trial. Finally, we would like to express our thanks to the staff of National Taiwan University Hospital-Statistical Consulting Unit (NTUH-SCU) for statistical consultation and analyses.

## Author Contributions

**Conceptualization:** Bing Min Tsai, Chiang Wen-Chu.

**Data curation:** Bing Min Tsai, Jen-Tang Sun, Ming-Ju Hsieh, Yu-You Lin.

**Formal analysis:** Bing Min Tsai, Jen-Tang Sun, Chiang Wen-Chu.

**Funding acquisition:** Matthew Huei-Ming Ma, Chiang Wen-Chu.

**Investigation:** Bing Min Tsai, Yu-You Lin, Chiang Wen-Chu.

**Methodology:** Ming-Ju Hsieh, Chiang Wen-Chu.

**Project administration:** Tsung-Chi Kao, Lee-Wei Chen, Chiang Wen-Chu.

**Resources:** Lee-Wei Chen, Matthew Huei-Ming Ma.

**Supervision:** Jen-Tang Sun, Ming-Ju Hsieh, Tsung-Chi Kao, Lee-Wei Chen, Matthew Huei-Ming Ma.

**Visualization:** Jen-Tang Sun, Chiang Wen-Chu.

**Writing – original draft:** Bing Min Tsai.

**Writing – review & editing:** Chiang Wen-Chu.

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
