## [Decision Letter · Decision Letter 0]

20 May 2020

PONE-D-20-07990

Optimal Paramedic Numbers in Resuscitation of Patients with Out-of-hospital Cardiac Arrest: A Randomized Controlled Study in a Simulation Setting

PLOS ONE

Dear Dr. WEN-CHU,

Thank you for submitting your manuscript to PLOS ONE. After careful consideration, we feel that it has merit but does not fully meet PLOS ONE’s publication criteria as it currently stands. Therefore, we invite you to submit a revised version of the manuscript that addresses the points raised during the review process.

We would appreciate receiving your revised manuscript by Jul 04 2020 11:59PM. To enhance the reproducibility of your results, we recommend that if applicable you deposit your laboratory protocols in protocols.io, where a protocol can be assigned its own identifier (DOI) such that it can be cited independently in the future. For instructions see: http://journals.plos.org/plosone/s/submission-guidelines#loc-laboratory-protocols

We look forward to receiving your revised manuscript.

Kind regards,

Corstiaan den Uil

Academic Editor

PLOS ONE

2. Please include your tables as part of your main manuscript and remove the individual files. Please note that supplementary tables (should remain/ be uploaded) as separate "supporting information" files

3. We note you have included a table to which you do not refer in the text of your manuscript. Please ensure that you refer to Table 2 in your text; if accepted, production will need this reference to link the reader to the Table.

Reviewers' comments:

Reviewer's Responses to Questions

**Comments to the Author**

1. Is the manuscript technically sound, and do the data support the conclusions?

Reviewer #1: Yes

Reviewer #2: Yes

2. Has the statistical analysis been performed appropriately and rigorously? 

Reviewer #1: Yes

Reviewer #2: No

3. Have the authors made all data underlying the findings in their manuscript fully available?

Reviewer #1: Yes

Reviewer #2: Yes

4. Is the manuscript presented in an intelligible fashion and written in standard English?

Reviewer #1: Yes

Reviewer #2: Yes

5. Review Comments to the Author

Reviewer #1: Thank you for giving me the opportunity to review your article. I could read this article with interest, however I felt some concern about your study.

The author mention the more crow reduce the time of intubation, and injection. However, there are many reports which do not recommend intubation and recommend compression only CPR. Therefore, this report has no worth for the investigators who believe such reports.

What kind of guideline did this study follow? RCA?

In this study, the author recommend the best cure size is 5. However, transportation is also the important work for the emergency medical service. In practice, can 5 crows work in an ambulance?

Reviewer #2: This study investigated the effect of paramedic crew size in the resuscitation of patients with out-of-hospital cardiac arrest (OHCA) remains inconclusive. The hypothesis was teams with a larger crew size have better resuscitation performance including chest compression fraction (CCF), advanced life support (ALS), and teamwork performance than those with a smaller crew size. The study was well designed with strong point as a randomized one.

Below are some comments:

1. The primary endpoint, chest compression fraction (CCF) is essentially continuous data with interval scale at the very least. And this has been used to power the study. However, the test statistics for the primary endpoint is based on non-parametric method, Kruskal-Wallis instead of 1-way ANOVA. Is there any particular reasons for using this non parametric approach? As it is commonly known, the use of non-parametric method may reduce the power to detect the intended meaningful difference stated in the hypothesis

2. There is not any plot showing the distribution of the primary endpoint chest compression fraction (CCF) among the five groups. This is an important plot that will enable us to check a few things for example the data distribution and hence whether assumption on normality is fulfilled.

3. In the statistical analysis section, it is mentioned about ‘For all non-parametric continuous data, … ‘. It is not defined what it means by non—parametric continuous data. What is the criteria for non-parametric?

4. For the Kaplan-Meier method used to compare the time-to-accomplishment of ALS intervention (i.e. first-dose of epinephrine and success of endotracheal intubation), there are 5 groups to compare yet only a single p-value is reported there. What exactly this single p-value represent? Have you done any further analyses to do pairwise comparisons of the KM curves? Have any adjustment of the p-values made?

5. In the same question as in (4) is also for time to finish intubation

6. For the primary endpoint CCF, how is the result if using ANOVA with the obtained/observed CCF instead of the ranks? With this approach, is it possible to do further analyses of pairwise comparisons between groups?

6. PLOS authors have the option to publish the peer review history of their article (what does this mean?). If published, this will include your full peer review and any attached files.

Reviewer #1: Yes: Shuichi Hagiwara

Reviewer #2: No

---

## [Author Response · Author response to Decision Letter 0]

10 Jun 2020

Review Comments to the Author

Reviewer #1: 

Thank you for giving me the opportunity to review your article. I could read this article with interest, however I felt some concern about your study.

The author mention the more crow reduce the time of intubation, and injection. However, there are many reports which do not recommend intubation and recommend compression only CPR. Therefore, this report has no worth for the investigators who believe such reports.

Response from the authors: 

Thank you for your comments. 

The effect of prehospital advanced life support (ALS), including endotracheal intubation and intravenous epinephrine on the outcome of patients with out-of-hospital cardiac arrest (OHCA), remained inconclusive in literature. However, there are many studies here in Taiwan showing the positive association between ALS and OHCA patient outcomes (reference 1, 2, 3 beneath), and that’s why we have to analyze the effect of crew size on these interventions. However, we do agree with your concern and will add it in the revised manuscript. 

Reference1, 2, 3

1. The Effect of Successful Intubation on Patient Outcomes after Out-of-Hospital Cardiac Arrest in Taipei. Chiang WC, Hsieh MJ, Chu HL, et al. Annals Emergency Medicine. 2018; 71(3):387-396.

2. The effect of the number and level of emergency medical technicians on patient outcomes following out of hospital cardiac arrest in Taipei. Sun JT, Chiang WC, Hsieh MJ, et al. Resuscitation. 2018;122:48-53

3. Prehospital intravenous epinephrine may boost survival of patients with traumatic cardiac arrest: a retrospective cohort study. Chiang WC, Chen SY, Ko PCI et al. Scandinavian Journal of Trauma, Resuscitation and Emergency Medicine 2015; 23:102-108.

Changes made (Please refer to the marked version):

Page 14, Line 8: 

Finally, this study examined the optimal crew size based on the CCF and prehospital ALS (i.e. endotracheal intubation and intravenous epinephrine). The results may not be applicable to those EMS systems without the recommendation of prehospital ALS in the field.

What kind of guideline did this study follow? RCA?

Response from the authors: 

Thank you.

Paramedics here in Taiwan resuscitate the OHCA patients in the prehospital setting following the recommendations of ILCOR (International Liaison Committee on Resuscitation), but the timing of applying a mechanical CPR device during resuscitation has not been uniformly regulated in the EMS protocol of New Taipei City. 

We have clarified this in the revised manuscript.

Changes made (Please refer to the marked version):

Page 9, Paragraph 2: 

Paramedics here in Taiwan resuscitate the OHCA patients in the prehospital setting following the recommendations of ILCOR (International Liaison Committee on Resuscitation), but the timing of applying a mechanical CPR device during resuscitation has not been uniformly regulated in the EMS protocol of New Taipei City. Hence, the time for deployment of M-CPR device was decided by every team themselves.

In this study, the author recommend the best cure size is 5. However, transportation is also the important work for the emergency medical service. In practice, can 5 crows work in an ambulance?

Response from the authors: 

The goal of this investigation aimed to help determine the best crew size for resuscitation of an OHCA patient in the field before ambulance transportation, thus we designed a simulation-based randomized controlled trial with cardiac arrest scenario equipped with widely adopted mechanical CPR to test different crew sizes. Based on the local EMS protocol, we will dispatch the 2nd crew to the scene in case of encountering an OHCA patient, but the transportation is still with only 2 paramedics in the ambulance patient compartment, not 5. 

Thank you for your comments. We have clarified this in the revised manuscript.

Changes made (Please refer to the marked version):

Page 6, Line 17-18: 

To help determine the best crew size for resuscitation of an OHCA patient in the field before ambulance transportation, we designed a simulation-based randomized controlled trial…

Reviewer #2: 

This study investigated the effect of paramedic crew size in the resuscitation of patients with out-of-hospital cardiac arrest (OHCA) remains inconclusive. The hypothesis was teams with a larger crew size have better resuscitation performance including chest compression fraction (CCF), advanced life support (ALS), and teamwork performance than those with a smaller crew size. The study was well designed with strong point as a randomized one.

Below are some comments:

1. The primary endpoint, chest compression fraction (CCF) is essentially continuous data with interval scale at the very least. And this has been used to power the study. However, the test statistics for the primary endpoint is based on non-parametric method, Kruskal-Wallis instead of 1-way ANOVA. Is there any particular reasons for using this non parametric approach? As it is commonly known, the use of non-parametric method may reduce the power to detect the intended meaningful difference stated in the hypothesis

Response from the authors: 

Thank you for your comments. We used the Kruskal-Wallis method for our primary end point because our sample size is relatively small, only 7 outcome data in each crew-size group.

We have clarified this in the revised manuscript.

Changes made (Please refer to the marked version):

Page 11, line 12-14:

… the Kruskal-Wallis method was used to examine significance because of sample size limitation (i.e. only 7 outcome data in each crew-size group).

2. There is not any plot showing the distribution of the primary endpoint chest compression fraction (CCF) among the five groups. This is an important plot that will enable us to check a few things for example the data distribution and hence whether assumption on normality is fulfilled.

Response from the authors: 

Thank you for your comments to make our manuscript more precise. 

It is important to plot the distribution of the primary endpoint. 

We’ve added scatter plots for primary and secondary outcomes in the revised manuscript.

Changes made (Please refer to the marked version):

Page 12, Line 16: The scatter plot of CCF data was shown in Appendix C and D.

Appendix Figure C: The scatter plot of primary endpoint (overall chest compression fraction, CCF) 

Appendix Figure D: The scatter plot of chest compression fraction (CCF) during manual period (hand compression) and mechanical period (machine compression).

3. In the statistical analysis section, it is mentioned about ‘For all non-parametric continuous data, … ‘. It is not defined what it means by non—parametric continuous data. What is the criteria for non-parametric?

Response from the authors: 

Thank you for reminding us of the misused statistical term. 

We have made a correction.

The criteria for non-parametric in this study was its sample size limitation. Because there was only 7 outcome data in each crew-size group, the assumption on normality is not fulfilled.

Changes made (Please refer to the marked version):

Page11, line11:

(misused term deleted)...The Kruskal-Wallis method was used to examine significance for sample size limitation (i.e. only 7 outcome data in each crew-size group), not fulfilling the assumption of normality.

4. For the Kaplan-Meier method used to compare the time-to-accomplishment of ALS intervention (i.e. first-dose of epinephrine and success of endotracheal intubation), there are 5 groups to compare yet only a single p-value is reported there. What exactly this single p-value represent? Have you done any further analyses to do pairwise comparisons of the KM curves? Have any adjustment of the p-values made?

Response from the authors: 

Thank you for your comments. 

The single p-value reported here is the overall p-value. Statistically significant results indicate that not all of the group means are equal.

Per your suggestions, we have consulted the staff of National Taiwan University Hospital-Statistical Consulting Unit (NTUH-SCU), and performed post hoc tests to do pairwise comparisons with the adjusted level of the p-value of the KM curves. The results were reported on the revised figure 3A&3B. 

Thank you for making our manuscript more informative to the readers.

Changes made (Please refer to the marked version):

Page 12, line 1: 

Post hoc tests were performed to do pairwise comparisons for the primary endpoint (CCF) and Kaplan-Meier curves, with the adjusted level of the p-value for multiple comparisons.

Figure. 3A & 3B (figure tag: results of overall p-value and post hoc test) 

5. In the same question as in (4) is also for time to finish intubation

Response from the authors: 

Thank you for the suggestions.

Changes made (Please refer to the marked version):

Page 12, line 1: 

Post hoc tests were performed to do pairwise comparisons for the primary endpoint (CCF) and Kaplan-Meier curves, with the adjusted level of the p-value for multiple comparisons.

Figure. 3A & 3B (figure tag: results of overall p-value and post hoc test) 

6. For the primary endpoint CCF, how is the result if using ANOVA with the obtained/observed CCF instead of the ranks? With this approach, is it possible to do further analyses of pairwise comparisons between groups?

Response from the authors: 

Thank you for your comments. We have made changes accordingly. 

Per your suggestions, we performed post hoc tests with the CCF. 

The results were reported on the revised Table 2. 

Thank you for making our manuscript more informative to the readers.

Changes made (Please refer to the marked version):

Revised Table. 2 (with a new column “post hoc tests”)

---

## [Editor Report · Decision Letter 1]

15 Jun 2020

Optimal Paramedic Numbers in Resuscitation of Patients with Out-of-hospital Cardiac Arrest: A Randomized Controlled Study in a Simulation Setting

PONE-D-20-07990R1

Dear Dr. WEN-CHU,

We’re pleased to inform you that your manuscript has been judged scientifically suitable for publication and will be formally accepted for publication once it meets all outstanding technical requirements.

Kind regards,

Corstiaan den Uil

Academic Editor

PLOS ONE
---

## [Editor Report · Acceptance letter]

18 Jun 2020

PONE-D-20-07990R1 

Optimal Paramedic Numbers in Resuscitation of Patients with Out-of-hospital Cardiac Arrest: A Randomized Controlled Study in a Simulation Setting 

Dear Dr. WEN-CHU:

I'm pleased to inform you that your manuscript has been deemed suitable for publication in PLOS ONE. Congratulations! Your manuscript is now with our production department. 

Kind regards, 

on behalf of

Dr. Corstiaan den Uil 

Academic Editor

PLOS ONE